# Sampling Trade-Offs in Duty-Cycled Systems for Air Quality Low-Cost Sensors

**DOI:** 10.3390/s22103964

**Published:** 2022-05-23

**Authors:** Pau Ferrer-Cid, Julio Garcia-Calvete, Aina Main-Nadal, Zhe Ye, Jose M. Barcelo-Ordinas, Jorge Garcia-Vidal

**Affiliations:** 1Computer Architecture Department, Universitat Politècnica de Catalunya, 08034 Barcelona, Spain; pau.ferrer.cid@upc.edu (P.F.-C.); aina.main@est.fib.upc.edu (A.M.-N.); zhe.ye@estudiantat.upc.edu (Z.Y.); jorge.garcia@upc.edu (J.G.-V.); 2Independent Researcher, 08034 Barcelona, Spain; julio468@gmail.com

**Keywords:** air quality, low-cost sensors, sampling, sensor calibration, duty cycle

## Abstract

The use of low-cost sensors in conjunction with high-precision instrumentation for air pollution monitoring has shown promising results in recent years. One of the main challenges for these sensors has been the quality of their data, which is why the main efforts have focused on calibrating the sensors using machine learning techniques to improve the data quality. However, there is one aspect that has been overlooked, that is, these sensors are mounted on nodes that may have energy consumption restrictions if they are battery-powered. In this paper, we show the usual sensor data gathering process and we study the existing trade-offs between the sampling of such sensors, the quality of the sensor calibration, and the power consumption involved. To this end, we conduct experiments on prototype nodes measuring tropospheric ozone, nitrogen dioxide, and nitrogen monoxide at high frequency. The results show that the sensor sampling strategy directly affects the quality of the air pollution estimation and that each type of sensor may require different sampling strategies. In addition, duty cycles of 0.1 can be achieved when the sensors have response times in the order of two minutes, and duty cycles between 0.01 and 0.02 can be achieved when the sensor response times are negligible, calibrating with hourly reference values and maintaining a quality of calibrated data similar to when the node is connected to an uninterruptible power supply.

## 1. Introduction

In recent years, much effort has been devoted to investigating energy-saving mechanisms in the design of wireless sensor networks. Most of this effort has focused on the communications subsystem [1] and little attention has been paid to the energy consumption of the sensing subsystem, assuming most of the time that sampling is performed instantaneously and at a negligible energy cost. The major efforts made regarding the reduction of energy consumption in the sensing subsystem are in in-network aggregation techniques, in which the aggregation process is performed in the multi-hop network [2]. The case of low-cost air pollution sensors is a different scenario [3]. Low-cost sensors report aggregated data that are then fed to a machine learning model to produce air pollution estimates. The way these samples are taken and the aggregation strategy have a great impact on the energy consumption of the sensor subsystem.

An additional challenge of these sensor networks that measure air pollution is that since the sensors have not been calibrated by the manufacturer or, they have been calibrated in laboratory chambers, they need to be calibrated in the environmental conditions of the deployment site [4,5]. Just as the sampling, pre-processing and aggregation tasks are performed at the sensing node (edge computing), the calibration task is performed off-line in the cloud. This calibration task allows to predict pollution values from raw sensor values. To calibrate air pollution sensors in an uncontrolled environment, they must be placed next to reference instrumentation, and therefore with the same sampling frequency as that used by the reference instrumentation. This methodology is known as in-situ calibration [4,5]. A widely used technique for these sensors is to use government reference stations [6] which, when connected to an uninterruptible power supply, aggregate the samples and display them every $T_{\mathrm{ref}}$ (for example, reference stations that display O${}_{3}$, NO${}_{2}$ or PM${}_{x}$ in Europe take samples continuously, during at least 45 min every hour, and display them every hour, see European directive 2008/50/EC) [7,8,9,10,11,12,13,14]. Low-cost sensors can follow a similar strategy if they are continuously powered, taking high-frequency samples (in the air pollution monitoring field, where reference stations provide measurements every hour, 2.7 × 10${}^{-4}$ Hz, a sampling period of few seconds is considered high frequency) and aggregating them into the same reference data periodicity $T_{\mathrm{ref}}$. In fact, most studies assume a high sampling frequency without taking into account energy consumption constraints [7,8,14,15]. Thus, if the nodes are powered by batteries it is challenging to implement a duty cycle-based strategy. The reason is that many of these sensors have a response period that can be on the order of several minutes before a correct measurement can be made. Besides, different air pollution phenomena may require different data sampling frequencies, which may have an impact on data quality. Concas et al. [16] discuss the critical steps in the use of low-cost sensors for air quality monitoring, specifically mentioning the data pre-processing step, including sensor sampling and sample aggregation. Different works have discussed the relevance of the trade-off between sensor sampling and power consumption in air pollution monitoring networks [17,18,19,20,21,22].

The objective of this work is to analyze the sampling strategies implications in this type of air quality sensor network, where it is necessary to implement a duty cycle strategy that saves energy in the sensor subsystem while achieving the best quality of reported data, according to the chosen calibration method. We place special emphasis on the impact that sensor sampling strategy has on calibration quality, as well as the impact of the resulting duty cycle. To this end, we calibrate O${}_{3}$, NO${}_{2}$, and NO electrochemical sensors using different duty cycle strategies. The experiments are performed using data collected by an experimental IoT node called Captor, which measures O${}_{3}$, NO${}_{2}$, NO, temperature, and relative humidity, and whose hardware and software have been developed to sample intensively in order to be able to simulate several sampling strategies. Specifically, in this work we:Describe the prototype node used to simulate the duty cycle strategies;Perform a multiple linear regression, k-nearest neighbors, and support vector regression calibration of O${}_{3}$, NO${}_{2}$, and NO sensors, assuming high data availability;Simulate different sensor sampling strategies, showing their impact on the goodness-of-fit of the calibration, as well as the implications of the resulting duty cycles.

The different sections are organized as follows: Section 2 shows the related work. Section 3 describes the experimental node used in this research work. Then, Section 4 introduces the different pre-processing steps required for sensor calibration. Section 5 shows the different experiments performed. Finally, Section 6 presents the conclusions of the paper.

## 2. Related Work

*Low-cost sensors in air pollution:* the use of low-cost sensors for air pollution monitoring has been the subject of study during the last few years [6,23]. These sensors provide a cost-effective alternative to complementing measurements from high-cost government-deployed instrumentation. The low cost of these sensors leads to low data quality, therefore the calibration of low-cost sensors has been studied in depth during the last years in order to improve the quality of the data [12,15,16]. Studies have been carried out to verify whether low-cost sensors can obtain accurate measurements and whether they can be included in a regulated way for air quality monitoring [6,24,25]. The most widely used technique for improving the quality of low-cost sensor data has been in-situ calibration using machine learning techniques [4,5,12,26,27,28]. In-situ calibration consists of placing the sensor in a reference station and using the obtained sensor values and reference values to train a supervised machine learning model. Several studies have used linear models such as multiple linear regression to calibrate sensors of different gases [14,29]. Besides, non-linear techniques, such as k-nearest neighbors, support vector regression, random forest or neural networks, have also been used to calibrate sensors [8,9,10,14,25,30].

*Data aggregation and duty cycle in air pollution sensors:* historically, the subsensing system for energy saving in air pollution sensor networks has been given little consideration. However, data pre-processing in air pollution low-cost sensors has a major impact on power consumption and data quality. Concas et al. [16] survey analyzes the necessary pre-processing steps for air pollution low-cost sensors, including aspects such as the need to aggregate data to reduce the sampling rate and the calibration of sensors. One of the reasons for having to aggregate is the synchronization of sampling intervals to minimize cross-sensitivities. The problem of data aggregation in air pollution sensor networks is thus different from other sensor networks. The nodes are calibrated individually, and the aggregation is performed in the node’s sensing subsystem and not in the network, so the objective here is not to minimize the number of packets to be transmitted together in the network, but the energy consumption of the sensing subsystem while maintaining the quality of the calibrated data. This implies defining correctly how often the sensors’ signal is sampled to maintain good data quality. Table 1 shows works in which the authors sampled at different frequencies to obtain calibrated data with air pollution sensors, mostly with nodes continuously connected to power. For the case of air pollution sensors, the trade-off between sensor sampling and node power consumption has been studied, but node calibration has not been taken into account, using already calibrated sensors [17,19,20,21,22]. Becnel et al. [18] propose a low-cost pollution monitoring station for airborne PM, temperature, relative humidity, light intensity, carbon monoxide, nitrogen oxide, that performs a duty cycle scheme to save energy when the node operates as a mobile node. However, the calibration analysis is performed only for the case of the node connected to an uninterruptible power supply. Thus, there remains a relevant aspect to be studied in low-cost sensors, which is the impact of sensor sampling on calibration and subsequent air pollution estimation.

*Our work:* As a summary, related works show how recently there has been an increasing interest in investigating the impact of duty cycle schemes on the sensing subsystem to reduce power consumption while maintaining data quality. Most works assume that the sensors have already been calibrated, and investigate the relationship between duty cycle techniques and energy consumption, mainly with PM${}_{2.5}$ sensors [17,19,20,21,22]. Few papers consider the impact of reduced sampling frequency on pre-processing steps such as data aggregation and sensor calibration. In this paper, we study the impact of the sampling frequency of air quality sensors on the calibration quality and duty cycle. In this way, we provide valuable information for professionals who want to build a node to measure air pollution and require energy savings in the sensor subsystem. Our work differs from the state-of-the-art in that most research investigating the trade-off between power consumption and frequency sampling assume that the sensors are already calibrated. Moreover, these works mostly investigate PM sensors. In our work, we consider other gas sensors, such as O${}_{3}$, NO${}_{2}$, and NO, and the trade-off between power consumption, data quality, and sampling frequency in sensor calibration.

## 3. Sensing Node: The Captor Node

To carry out the study, we developed a data capture node, called Captor, whose purpose is to attain maximum flexibility and scalability to integrate sensors, and to be able to experiment with different duty cycle strategies and sampling frequencies.

### 3.1. CAPTOR Node

The Captor node is built based on an I2C bus, to which different sensing subsystems, and a master central node, which is in charge of managing a communication subsystem, are attached, see Figure 1. In the specific case of the data reported in this work, a gas monitoring shield designed to integrate two Alphasense sensors is used. The sensing board is built using an Arduino Nano microcontroller unit (MCU), which samples the data measured from two gas sensors, and a temperature and relative humidity sensor. Each gas sensor is supplied by the manufacturer with an individual sensor board [32]. The output of the individual sensor board is further amplified by a factor of ×2, in order to reduce quantifying errors, and sampled by the analog-to-digital converter of the Arduino Nano MCU. The sensor sampling frequency is set to 0.5 Hz. The data captured by the Arduino Nano MCU is periodically sent through the I2C bus to the central processing unit, based on a Raspberry Pi node, which also handles the communications subsystem. The node is designed to scale the number of sensors included by adding new sensing boards connected to the I2C bus.

### 3.2. Low-Cost Sensors

To carry out this work, Alphasense electrochemical gas sensors have been used. Specifically, we have used OX-B431 O${}_{3}$ sensors [33], NO2-B43F NO${}_{2}$ sensors [34], and NO-B4 NO sensors [35]. Although other technologies exist, electrochemical sensors are an inexpensive sensing technology that is widely used for air pollution in today’s low-cost sensor monitoring networks [7,9,11,13,31]. The B4 sensor family, designed for use in urban air fixed-site networks, is a 4-electrode (working, reference, counter, and auxiliary) electrochemical sensor with very low parts per billion (ppb) detection levels. Each sensor is provided with an individual sensor board [32] that requires 3.5 V to 6.4 V stable direct current supply with a consumption around 1 mA. To measure a pollutant, the individual sensor board sends two raw values to a 10-bit analog-to-digital converter: the working electrode (WE) is the reduction or oxidation site of the chosen gas species, and the auxiliary electrode (AE) is used to correct for zero current changes [13]. The final raw signal is obtained by subtracting the raw working and auxiliary values produced by the analog-to-digital converter, S=WE−AE, or by feeding both parameter values, WE and AE, to the machine learning algorithm as separate features. The O${}_{3}$ sensor is a special case since the working electrode measures O${}_{3}$ and NO${}_{2}$ simultaneously. This means that to obtain O${}_{3}$ it is necessary to use a pair of OX-B431 O${}_{3}$ and NO2-B43F NO${}_{2}$ sensors. Furthermore, it should be noted that the manufacturer of these sensors indicates that they have a response time $T_{r}$ of less than 80 s for O${}_{3}$ and NO${}_{2}$ [33,34] and 45 s for NO [35]. This is important for the implementation of strategies that minimize the duty cycle of the sensing subsystem, since it is necessary, for example, to wait for at least 80 s for O${}_{3}$ and NO${}_{2}$ before obtaining valid measurements from the sensors if the sensing boards are switched off. Other sensors may have a response time in the order of milliseconds, which is negligible in the case of duty cycles of higher orders of magnitude.

### 3.3. Datasets

This section describes the datasets obtained from Captor node prototypes that are later used for the experiments in Section 5. Two Captor nodes were deployed during four months at a reference station in Palau Reial, Barcelona (Spain). Captor node labeled as 20001 mounted one Alphasense OX-B431 O${}_{3}$ sensor, one Alphasense NO2-B43F NO${}_{2}$ sensor, one Alphasense NO-B4 NO sensor, and a DHT-22 temperature and relative humidity sensor. The Captor node labeled as 20002 mounted one Alphasense OX-B431 O${}_{3}$ sensor, one Alphasense NO2-B43F NO${}_{2}$ sensor, and a DHT-22 temperature and relative humidity sensor. These datasets allow the study of sampling policies, given their high temporal resolution, and the study of calibration methods for three different air pollutants using electrochemical sensors. Given the availability of high-frequency measurements (0.5 Hz), different sensor sampling policies can be simulated by subsampling these datasets (we emphasize that the sensing boards were not put to sleep, and therefore, the effect that turning the sensor on and off may have on aging or sample quality has not been studied).

Each of the low-cost electrochemical sensors provides measurements from the working electrode and auxiliary electrode in analog-to-digital converter units. The temperature sensor collects measurements in degrees Celsius (°C), and the relative humidity sensor collects measurements in percent humidity (%). Table 2 summarizes the different sensor data used in the experiments. To simulate what a real sensor deployment would be like, the two nodes were placed at a reference station for 4 months, from 2021/01/15 to 2021/05/15. In this way, the datasets have a length representative of a real monitoring campaign and reference values are available to check the quality of the data. The average concentrations measured by the reference station at Palau Reial (Barcelona) from 2021/01/15 to 2021/05/15 are 57.46, 19.87 and 4.28 $\mu \mathrm{gr}/m^{3}$ for O${}_{3}$, NO${}_{2}$ and NO, with standard deviations of 23.79, 15.31 and 11.74 $\mu \mathrm{gr}/m^{3}$ respectively. As shown in Section 5.1, the NO${}_{2}$ and NO present important concentration peaks above 100 $\mu \mathrm{gr}/m^{3}$. Reference station’s values are available hourly, so the reference data period $T_{\mathrm{ref}}$ is equal to one hour. The reference station’s data can be downloaded from the government’s open data web [36], while the raw Captor sensory data have been made public on Zenodo’s website [37].

## 4. Sensor Data Gathering Pipeline

In this section, we show the different steps required to obtain air pollution estimates from low-cost sensors. The whole sensor data gathering process can be divided into two stages; sensor data pre-processing, and the estimation of pollutant concentrations using a supervised machine learning algorithm. All data pre-processing can be performed at the sensing node so that only measurements synchronized with the reference values are transmitted to the cloud, and the subsequent estimation of pollutant data is performed off-line. It is important to note that, to estimate the final pollutant concentration, the machine learning algorithm, as described in Section 4.2, uses a combination of samples taken by several sensors of the same node. This edge computing approach significantly reduces the amount of data transmission required, since only the aggregation of collected data is sent to the cloud. From now on, we focus on how to reduce power consumption in the sensing subsystem when taking samples. Figure 2 illustrates all the required pre-processing steps to be able to perform the sensor data gathering procedure.

In this specific paradigm, the value of the reference station produced every hour is the aggregation of different samples taken during that hour at a frequency which we assume to be higher than the Nyquist frequency corresponding to the time variation of the measured phenomenon. If the sampling frequency of the low-cost sensor is also higher than the Nyquist frequency, we can expect the errors in the calibration process to be essentially independent of the sampling frequency of the low-cost sensor. However, if the frequency sampling of the low-cost sensor falls below the Nyquist frequency, we can expect this undersampling to introduce an additional source of error in the calibration process that can have a large impact on the accuracy of the measured values during sensor operation.

### 4.1. Pre-Processing

Data pre-processing has a big impact on the subsequent representation of the data. As mentioned above, having the data synchronized with reference stations, in the environment where the node will be deployed, allows us to calibrate the sensors and to detect drifts, aging or outliers [38,39,40], which will lead to a recalibration of the sensor. Specifically, we divide the pre-processing operation into three stages; the sampling of the sensor, the filtering of the collected samples, and the aggregation of these samples. The process of taking measurements in air pollution sensors is as follows (Figure 2): First, the node retrieves a value of the sensor every $T_{\mathrm{sen}}$ seconds. For this value to be representative, it may be necessary to wait for a sensor response time $T_{r}$, take a sequence of $N_{s}$ measurements, apply a filtering algorithm to remove outliers and smooth the measurements, and finally aggregate them into the sample $T_{\mathrm{sen}}$. The microcontroller can go into sleep mode until it has to collect samples again and switch off the sensing board if necessary. The node manages an array of sensors, each with its own electronic board, whereby the node reports a vector, each $T_{\mathrm{sen}}$, containing the air pollution sensor values (e.g., NO${}_{2}$, O${}_{3}$, NO) and environmental values (e.g., temperature and relative humidity). Two strategies are possible: (i) a packet is generated with the sample vector every $T_{\mathrm{sen}}$; or (ii) if energy savings are desired in the communication subsystem and the application only needs values every $T_{\mathrm{ref}}$, a second aggregation is carried out and transmitted every $T_{\mathrm{ref}}$.

#### 4.1.1. Sensor Sampling

At this stage, the $N_{s}$ samples that are part of the representative sample $T_{\mathrm{sen}}$ are taken. We focus on taking samples from a single sensor. In the case of having an array of sensors, the microcontroller can run the sampling process in parallel, activating all the sensor boards simultaneously and polling them with a round robin strategy. Besides, since different sensors are attached to different sensing boards, a specific sampling strategy per sensor can be designed.

To obtain the value at instant $T_{\mathrm{sen}}$ the microcontroller wakes up the sensor board and takes $N_{s}$ consecutive samples (Figure 3). In this case, the duty cycle is $\left(N_{s}T_{s}\right)/T_{\mathrm{sen}}$. However, there are air pollution sensors, for example, the ones used in this article that have a response time of $T_{r}$, so it is necessary to wait for $T_{r}$ before collecting valid measurements. Indeed, this response time may vary from one sensing technology to another, and it can be seen as a user-defined parameter to specify the amount of time to wait before collecting a measure to prevent the collection of incorrect measurements. Summarizing, the microcontroller wakes up the sensor board, waits for a time $T_{r}$ and then takes $N_{s}$ consecutive samples to build the value $T_{\mathrm{sen}}$ and turns off the sensor board. In this general case, the duty cycle DC is given by:(1)$$\begin{array}{cc} \mathrm{DC}= & \;\frac{T_{\mathrm{on}}}{T_{\mathrm{sen}}}\\ T_{\mathrm{on}}= & \;T_{r}+\left(N_{s}\cdot T_{s}\right)\\ T_{\mathrm{on}}\leq & \;T_{\mathrm{sen}}. \end{array}$$

The number of samples $N_{s}$ that make up the value generated by $T_{\mathrm{sen}}$ impacts the duty cycle of the sensing subsystem and the quality of the data estimated by the machine learning algorithm. The adjustment of the value of $T_{\mathrm{sen}}$ has an impact on the number of packets to transmit, on the duty cycle of the sensor subsystem, and on the quality of the value estimated by the machine learning algorithm. Table 3 summarizes the different sampling parameters used throughout the paper.

#### 4.1.2. Filtering

Once $N_{s}$ sensor samples have been collected for each sensing node’s sampling period $T_{\mathrm{sen}}$, these must be filtered in order to remove outliers and smooth the data. One of the most common techniques for removing outliers in sensory data is the use of the z-score [14]. Other signal filtering techniques (e.g., moving average) can be useful to eliminate abrupt changes in the signal and smooth the data tendency. For instance, Mijling et al. [7] eliminate samples that deviate a given percentage from the sample mean. The filtering process is necessary because the signals measured by the sensors are noisy and tend to produce outliers, in which case the subsequent aggregation would be affected and, consequently, the quality of the estimated data would be degraded. The computational cost of this filtering is minimal, as it requires only a few operations on the data collected every $T_{\mathrm{sen}}$ period. Moreover, following an edge computing approach, this filtering is calculated at the node itself. In the experiments Section 5, we use the z-score as filtering technique and we remove extreme values.

#### 4.1.3. Aggregation

In the aggregation stage, the sensor data, after filtering, are aggregated into a single measurement period $T_{\mathrm{sen}}$. The most common statistics used for the aggregation are the sample mean and median. However, these statistics require a certain number of samples in order to not be biased, so after the filtering step, it is important to check whether the resulting number of samples in a period is large enough for averaging, if not, the resulting mean may not be representative and the sample is discarded producing a gap. This aggregation has a minimal computational cost and is computed at the node itself before the packets are transmitted.

The measured value is now included in a vector of measurements from all sensors on the node, and can be transmitted to the cloud where the machine learning algorithm can estimate the pollution value with granularity $T_{\mathrm{sen}}$. The reference stations, being connected to power supply, usually take continuous $T_{\mathrm{sen}}$ values and aggregate those values into hourly values ($T_{\mathrm{ref}}=\;1\;h$), which are the ones displayed in the applications. If we want to save energy in the communications subsystem, we can do a second aggregation with the $T_{\mathrm{sen}}$ values to match the values of the reference stations $T_{\mathrm{ref}}$ values. This allows having a heterogeneous network of reference stations and low-cost sensor nodes that can spatially measure a pollutant in an area as the two types of nodes have the same time granularity.

However, nodes with low-cost sensors that have a response time of more than a minute and that also use batteries and implement a duty cycle to save energy in the sensing subsystem, will not be able to produce $T_{\mathrm{sen}}$ values in the same way as reference stations. In this work, we investigate different ways to implement such a duty cycle, from a strategy that tries to mimic as much as possible, given the constraints of the low-cost sensors, a reference station, to a more aggressive strategy that although it does not follow the same dynamics as a reference station, produces an aggregated value at the $T_{\mathrm{ref}}$ instant saving the maximum energy. The most aggressive strategy that can save more energy is to consider that the value taken every $T_{\mathrm{ref}}$ only considers a $T_{s}$ value. Nevertheless, this value may be unrepresentative of the physical phenomenon during the whole $T_{\mathrm{ref}}$ interval. The less aggressive strategy, which saves less energy and better mimics the reference station is to consider that during $T_{\mathrm{ref}}$ we have as uniformly as possible $N_{s}\left\lfloor T_{\mathrm{ref}}/T_{\mathrm{sen}}\right\rfloor$ measurements. In the results Section 5, we review the two strategies, where the $N_{s}$ samples are taken uniformly in $T_{\mathrm{sen}}$ or are taken consecutively in $T_{\mathrm{sen}}$ assuming a sensor response interval $T_{r}$, showing their feasibility and implications in terms of data quality and power consumption. In the experiments Section 5, we use the mean as aggregation technique.

### 4.2. Machine Learning-Based Sensor Calibration

To estimate the values of an air pollutant we use calibration techniques based on machine learning. To this end, we need the sensing node to produce values with the same granularity $T_{\mathrm{ref}}$ as the reference station in order to compute the coefficients or hyperparameters of the machine learning algorithm. The process of obtaining these coefficients or hyperparameters is called sensor calibration [4,5]. Several machine learning techniques [8,9,10,11,25] have been used to improve the accuracy of the calibration and to obtain air pollution measurements from raw sensor values. Ultimately, sensor calibration is reduced to a supervised problem where we have the pairs ${\left\{x_{i},y_{i}\right\}}_{i=1}^{N}$ where $x_{i}\in R^{P}$ are the sensors values, where *P* is the number of sensors included in the calibration, and $y_{i}\in R$ are the corresponding reference values. As an example, for calibrating an electrochemical Alphasense OX-B341 O${}_{3}$ sensor, we need the raw values from an Alphasense OX-B341 sensor, an Alphasense NO2-B43F NO${}_{2}$ sensor, a temperature sensor, and a relative humidity sensor [7,8,11,13,28]. That means that the vector x has dimension four. Given these data, we can formulate the following problem:(2)$$y_{i}=f\left(x_{i}\right)+\epsilon_{i};\;\;\forall i=1,..,N,$$
where f(·) is the function to be determined by machine learning, and $\epsilon_{i}$ is the error assumed to be independent and identically distributed. There are different algorithms to estimate the function f(·), among which we have the multiple linear regression, and nonlinear models such as k-nearest neighbors, random forests, support vector regression or artificial neural networks [8,25,26,28]. We have decided to use three state-of-the-art in-situ calibration models: multiple linear regression, k-nearest neighbors, and support vector regression. Thus, we use three machine learning models belonging to different classes of machine learning models; linear methods, instance-based methods, and kernel methods. The multiple linear regression assumes that the reference values, $y_{i}$, vary linearly depending on the sensor values, so (Equation 2) can be rewritten as:(3)$$y_{i}=\beta_{0}+\beta^{T}x_{i}+\epsilon_{i};\;\;\forall i=1,..,N,$$
where $\left[\beta_{0}\;\beta \right]$ is the vector of coefficients to be found by solving the problem by least squares.

The k-nearest neighbors model obtains the prediction $\hat{f}\left(x\right)$ for a data instance x by finding the k-closest training instances $N_{k}\left(x\right)$, using a distance metric $d(x,x_{i})$, and averages the response values $y_{i}$ of these instances:(4)$$\hat{f}\left(x\right)=\frac{1}{k}\sum_{i\in N_{k}\left(x\right)}y_{i}.$$

The support vector regression makes use of the representer theorem and defines the estimation of the function f(·) as:(5)$$\hat{f}\left(x\right)=\sum_{i=1}^{N}\left({\hat{\alpha }}_{i}^{*}-{\hat{\alpha }}_{i}\right)K\left(x,x_{i}\right)+b,$$
where $K:R^{P}\times R^{P}\to R$ is the kernel function, ${\left\{x_{i}\right\}}_{i=1}^{N}$ are the set of training instances, and ${\hat{\alpha }}_{i}^{*}$,${\hat{\alpha }}_{i}$ and *b* are the parameters to be estimated via convex optimization. The k-nearest neighbors and support vector regression models have different hyperparameters that need to be calculated. These values are selected using a cross-validation procedure performing a grid search over different possible values. For further information on the application of these supervised machine learning models for in-situ sensor calibration refer to [8,25,26,28].

## 5. Results

In this section, we perform two different experiments to evaluate the impact of the sampling strategy on the in-situ calibration of low-cost sensors using the datasets described in Section 3.3:We perform sensor in-situ calibration for the O${}_{3}$, NO${}_{2}$ and NO sensors, using the best subsets of sensors found by forward stepwise feature selection using 10-fold cross-validation (CV). We perform this experiment using the raw sensors’ signals ($T_{\mathrm{sen}}=\;2\;s$), therefore assuming no energy consumption restrictions. We compare three in-situ calibration models: multiple linear regression, k-nearest neighbors and support vector regression;We investigate the impact of the sensor sampling parameters on the sensor calibration accuracy and power consumption. To this end, we compare different sampling periods for the sensing node $T_{\mathrm{sen}}$, as well as different numbers of samples collected in these periods $N_{s}$. We also consider sensor response periods. To carry out this experiment, we use the raw two-second signals from the sensors and simulate the different sampling settings by subsampling these raw signals. A 10-fold CV is performed for every sampling setting, discussing the resulting goodness-of-fit metrics, duty cycles, and power consumption implications.

To evaluate each one of the experiments, 75% of the dataset is used as a training set, and the remaining 25% is used as a testing set. The different datasets are shuffled so that the training conditions are representative of the testing, avoiding out-of-date and inaccurate calibration models [5,8,16], and the different experiments can be evaluated without the effects of the changing environmental conditions.

### 5.1. Machine Learning-Based Calibration

Given the aggregated sensor data, synchronized with the reference data, sensor calibration can be performed using machine learning techniques. This is the ultimate step in obtaining air pollution estimates using low-cost sensors. In this section, we use the raw sensors’ signals, so at the highest data availability ($T_{\mathrm{sen}}\;=\;2\;s$). Therefore, we show the best case of having the sensing node connected to an uninterruptible power supply collecting sensor measurements uniformly every two seconds. In this way, we can show the ability of the different sensors to predict the real air pollutant concentrations in the case of not having energy restrictions.

Since the data collection nodes have up to three sensors measuring different pollutants (O${}_{3}$, NO${}_{2}$, and NO), several sensors can be introduced into the calibration models to take advantage of the cross-sensitivities and correlations present. Therefore, we define the sensors to be introduced in the sensor calibration using forward stepwise feature selection using 10-fold CV. In this way, in the forward stepwise process the sensor that most significantly increases the cross-validation R${}^{2}$ is added each time until there is no significant improvement. The best subset of sensors found for the machine learning calibration for each sensor are shown in Table 4. Temperature and relative humidity sensors are included in the calibration since they are known to be correctors of environmental conditions for the O${}_{3}$, NO${}_{2}$ and NO sensors [7,11,13,26]. For instance, in the NO calibration case, the O${}_{3}$ sensor, which measures both O${}_{3}$ and NO${}_{2}$, is the sensor that introduces the biggest improvement apart from the NO sensor. From now on, the combination of sensors shown in the Table 4 is assumed in all calibration models.

Figure 4a shows the results of the calibration of the Captor 20001 O${}_{3}$ sensor using the multiple linear regression model. As can be seen, the test R${}^{2}$s is very good, specifically 0.97, with root-mean-square errors (RMSEs) of 4.40 $\mu \mathrm{gr}/m^{3}$ and 4.22 $\mu \mathrm{gr}/m^{3}$ in the case of the Captor 20002 O${}_{3}$ sensor. Figure 4b shows the calibration results for the Captor 20001 NO${}_{2}$ sensor, where the model obtains a testing R${}^{2}$ of 0.94 with RMSE of 3.85 $\mu \mathrm{gr}/m^{3}$, similarly Captor 20002 NO${}_{2}$ sensor achieves a testing R${}^{2}$ of 0.93, with an RMSE of 4.14 $\mu \mathrm{gr}/m^{3}$. Finally, Figure 4c shows the calibration for the Captor 20001 NO sensor, with a testing R${}^{2}$ of 0.91 and a testing RMSE of 3.26 $\mu \mathrm{gr}/m^{3}$. The calibration of NO is very dependent on the data used for calibration and testing, as NO has very abrupt peaks, it is important that there is data in the training that represents these peaks. Hence, the calibration goodness-of-fit depends on the pollution peaks observed during the training and testing periods. The nonlinear methods (support vector regression and k-nearest neighbors) performed similarly to the MLR given that the sensors’ responses are quite linear. In fact, the support vector regression obtained a R${}^{2}$ of 0.98 for both O${}_{3}$ sensors, R${}^{2}$ of 0.96 and 0.94 for the 20001 NO${}_{2}$ and 20002 NO${}_{2}$ sensors, and a R${}^{2}$ of 0.97 for the NO sensor, while the k-nearest neighbors obtained a R${}^{2}$ of 0.96 for both O${}_{3}$ sensors, R${}^{2}$ of 0.94 and 0.92 for the 20001 NO${}_{2}$ and 20002 NO${}_{2}$ sensors, and a R${}^{2}$ of 0.95 for the NO sensor.

### 5.2. Sensor Sampling Impact

In this experiment, we explore the impact of the sampling period $T_{\mathrm{sen}}$ and the number of samples collected $N_{s}$ on the model’s accuracy, and the implications of the resulting duty cycles with respect to the power consumption of the sensing node. The different sampling strategies are simulated by subsampling the raw sensors’ signals ($T_{\mathrm{sen}}\;=\;2\;s$). Since reference values are available hourly, the periods $T_{\mathrm{sen}}$ tested are less than or equal to one hour $T_{\mathrm{sen}}\;\leq \;1\;h$. We have obtained similar results for the three calibration models. For simplicity, we show the results using the MLR.

#### 5.2.1. Impact of Node Sampling Period $T_{\mathrm{sen}}$

The following experiment shows the result of calibrating the sensors by subsampling the measurements with different sampling periods $T_{\mathrm{sen}}$, from 2 s and 1 min to 60 min. Besides, we compare two strategies, one consisting of taking the $N_{s}$ samples uniformly over $T_{\mathrm{sen}}$, only possible when the sensor’s response time is small or negligible ($T_{r}\;\ll \;T_{\mathrm{sen}}$), and the other consisting of taking $N_{s}$ consecutive samples in $T_{\mathrm{sen}}$ after a large sensor response period $T_{r}$ that can be up to tens of seconds.

Figure 5a shows the average CV R${}^{2}$ and the confidence intervals for applying different periods $T_{\mathrm{sen}}$ and $N_{s}$ samples to the O${}_{3}$ calibration. As can be seen, in the case of sampling consecutively (solid lines), there is very little worsening of the performance from sampling every minute to sampling every 10 min. From this point, the R${}^{2}$ starts to decrease until it reaches an average CV R${}^{2}$ of 0.87. In the case of the 30 min and 60 min periods, it should be noted that the sensor is only being sampled twice or once (for $N_{s}\;=\;1$) per $T_{\mathrm{ref}}$, so the accuracy may decrease considerably. The difference between taking one, five, or ten samples is not significant until we sample every 30 to 60 min. However, as expected, when we take a few samples, e.g., $N_{s}\;=\;1$, the confidence intervals are worse, meaning that the quality of the calibration can exhibit variability when only sampling once. We note as an example, that in case of having $T_{\mathrm{sen}}\;=\;5\;\min$, the aggregation involves 12 values when calibrating at $T_{\mathrm{ref}}\;=\;60\;\min$. This indicates that in the case of frequent sampling (e.g., $T_{\mathrm{sen}}\;\leq \;5\;\min$) it is not necessary to take more than one sample, but when only sampling fewer times $T_{\mathrm{sen}}$ per reference period $T_{\mathrm{ref}}$, even if more than one sample is taken, the quality of the calibration is not maintained, since the subsequent aggregation involves samples, even if it has more samples in total, taken too consecutively in unrepresentative instants. For instance, if $T_{\mathrm{sen}}\;=\;30\;\min$ and $N_{s}\;=\;10$, there are 20 samples participating in the aggregation at instant $T_{\mathrm{ref}}$ (more than the 12 samples with $T_{\mathrm{sen}}\;=\;5\;\min$ and $N_{s}\;=\;1$) but they are less representative. In other words, it is better to sample fewer measurements more distributed over the period $T_{\mathrm{ref}}$, than to sample more measurements consecutively but fewer times at $T_{\mathrm{ref}}$.

Figure 5b shows a similar pattern for the NO${}_{2}$ sensor, but with a larger decrease in R${}^{2}$ for large sampling intervals. Indeed, the R${}^{2}$ is observed to remain almost constant for $T_{\mathrm{sen}}\;\leq \;10\;\min$, with values around 0.94. However, the worsening in this case is much greater than in the case of O${}_{3}$, since at $T_{\mathrm{sen}}\;=\;30\;\min$ the R${}^{2}$ is reduced to 0.86, and at $T_{\mathrm{sen}}\;=\;60\;\min$ to 0.75. Regarding the number of samples taken every period, it is seen that one sample is not significant enough, so this sampling setting works worse than the others for sampling intervals larger than 10 min. In addition, since NO${}_{2}$ is a less smooth signal than O${}_{3}$, with few samples per $T_{\mathrm{sen}}$ interval, there is greater variability, which explains the higher confidence interval values for $N_{s}\;=\;1$. Finally, for the uniform sampling approach, the same trend is observed for NO${}_{2}$ as for O${}_{3}$, where five to ten uniform samples over $T_{\mathrm{sen}}$ obtain very good data quality.

Figure 5c shows the results for the NO sensor calibration. In this case, NO is a phenomenon that naturally presents more abrupt changes in the measurements which causes the different calibrations to have large confidence intervals. The same decreasing trend is observed as for O${}_{3}$ and NO${}_{2}$ R${}^{2}$, where from $T_{\mathrm{sen}}\;=\;10\;\min$ the R${}^{2}$ starts to decrease, and it is also observed how the R${}^{2}$ strongly depends on the number of samples taken since the gap in performance between taking one sample and five is the largest of all three pollutants. The NO signal contains many peaks so even sampling $N_{s}$ samples in a row for an instant that does not pick up such peaks may be unrepresentative. Another effect is the uncertainty we will have in the measurements. In those cases where the number of samples is small, e.g., $N_{s}\;=\;1$, the confidence interval is very poor, precisely because of the high variability of the data. This interval improves when taking more samples, even if the measurement point is not very representative, and the R${}^{2}$ decreases, the confidence interval decreases. Therefore, in the case of signals with high variability and high bandwidth, it is logical to sample at more points. In the case of sampling uniformly, the same trend as for the two previous pollutants is observed, where taking at least five uniform samples is enough to maintain the highest data quality.

From all this, we can conclude that it is better to take more than one sample N${}_{s}>1$ if the sampling period is large ($T_{\mathrm{sen}}\;\geq \;10\;\min$), so that the aggregation is more representative. However, in the case of having a lower sampling period, fewer samples are enough to obtain a high R${}^{2}$ since the aggregation at each $T_{\mathrm{ref}}$ will contain enough samples to be representative. Moreover, different sensors may need different sampling strategies to maintain similar data quality in the prediction phase if energy savings are to be achieved. For instance, the performance gap between sampling five uniform samples for $T_{\mathrm{sen}}\;=\;30\;\min$ and five consecutive samples for $T_{\mathrm{sen}}\;=\;30\;\min$ is of 0.02 R${}^{2}$ in the O${}_{3}$ case, 0.05 R${}^{2}$ in the case of the NO${}_{2}$, and 0.08 R${}^{2}$ in the NO case.

#### 5.2.2. Impact of Duty Cycle DC: Data Quality

Now, we compare the data quality implications of the different sampling policies with respect to the duty cycles, which will have different energy consumption consequences. We explore two possible cases; negligible sensor response time $T_{r}\;\approx \;0$ and response time equal to two minutes $T_{r}\;=\;2\;\min$, given that data sensor responses may take up to 80 s, see Section 3.2. We assume that the time to turn on and off the sensing device is negligible. Figure 6a–c shows the R${}^{2}$ with respect to the duty cycle with negligible response time using MLR. Figure 6a shows the results for the O${}_{3}$ sensor, where it can be seen that since there is negligible sensor response time there is no penalty for turning on the node too many times. Indeed, for similar duty cycles, the strategy that takes one single sample frequently is able to achieve an R${}^{2}$ of 0.96, while the strategy that samples five measurements in a row achieves an R${}^{2}$ of 0.89 with a similar duty cycle. Sampling strategies that sample more than one measure have larger duty cycles since the sensing node needs to remain measuring more time. Regarding the uniform sampling strategy (dotted lines), it is observed that with $N_{s}\;=\;5$ samples it is able to achieve a very good goodness-of-fit at a very low duty cycle. However, this case is not representative at all, since the sensor’s response time will rarely be negligible and the presence of a sensor response time makes the strategy of sampling uniformly infeasible. Figure 6b shows the same results for the NO${}_{2}$ sensor, again the same results are seen, where for similar duty cycles the strategy that takes one single measure obtains an R${}^{2}$ of 0.93 while the strategy that takes five samples obtains an R${}^{2}$ of 0.75. That is, the setting $\{T_{\mathrm{sen}}\;=\;15\;\min$, $N_{s}\;=\;1\}$ works much better than the setting $\{T_{\mathrm{sen}}\;=\;60\;\min$, $N_{s}\;=\;5\}$ with a similar duty cycle. Figure 6c shows the results for the NO sensor, with the same pattern but with a worse average performance and larger variability because of the variability of this fast-changing phenomenon.

The results of the duty cycle for a 2 min sensor response time ($T_{r}\;=\;2\;\min$) and therefore with a $T_{\mathrm{sen}}\;>\;2\;\min$ are shown below. Here, only the results of the strategy that takes the measurements sequentially are shown, since with a sensor response time of two minutes it is no longer possible to take samples uniformly as the node would always be powered on. Recall that low duty cycles correspond to large $T_{\mathrm{sen}}$ and fewer samples taken in the interval $T_{\mathrm{ref}}$. This can be seen in Figure 7a–c, where lower R${}^{2}$ are obtained for low duty cycles. The coefficients of determination start to stabilize at DC=0.20 ($T_{\mathrm{sen}}\;=\;10\;\min$). Thus, a sampling period of about five or ten minutes guarantees the representativeness of the sampled data. When the physical phenomenon presents large variability, as in the case of NO, the confidence intervals are poor. However, large $T_{\mathrm{sen}}$ periods with one single sample introduce more variability, as observed in the confidence intervals of sampling strategies with $N_{s}\;=\;1$. In this case, it is better to take more samples, slightly increasing the duty cycle, since the sensor response time is the one that dominates the duty cycle.

#### 5.2.3. Impact of Duty Cycle DC: Power Consumption

The implications of a higher or lower duty cycle on the power consumption of a node depend on the different components used by the node. However, we can assume that the power consumption of the node is directly given by the duty cycle, as it is the ratio between the time it is powered on during the sampling period ($T_{\mathrm{on}}$) and the sampling period ($T_{\mathrm{sen}}$). Furthermore, depending on the consumption of the different components, one can decide to send only the microcontroller to sleep mode or send the microcontroller to sleep and switch off the sensors.

Now, looking at the results for the duty cycle with sensor response time, Figure 7a–c, it can be observed that with a duty cycle of about 0.10 the quality of the calibrations in two cases (O${}_{3}$ and NO${}_{2}$) are stable, introducing very little improvement at higher duty cycles rates. In the case of the NO sensor, a slightly higher duty cycle may be required (DC=0.15). This means that a calibration almost as good as when the node is always on (DC=1) can be obtained with a duty cycle about seven or ten times smaller (DC=0.15 and DC=0.10), therefore reducing the power consumption very significantly. Table 5 shows the average CV R${}^{2}$ for all the tested sensors and different duty cycles obtained (with sensor response time equal to two minutes) using different calibration models and sampling strategies; duty cycles equal to 1.0 ($\left\{T_{\mathrm{sen}}\;=\;2\;s,N_{s}\;=\;1\right\}$), 0.10 ($\left\{T_{\mathrm{sen}}\;=\;20\;\min,N_{s}\;=\;1\right\}$), and 0.03 ($\left\{T_{\mathrm{sen}}\;=\;60\;\min,N_{s}\;=\;1\right\}$). As it is observed, for duty cycles of 0.10 the sensor calibrations worsen by about 0.02–0.08 R${}^{2}$, in the worst case, the NO sensor drops from 0.90 to 0.82 R${}^{2}$. This means that the NO sensor may need a higher duty cycle, about 0.15, so that its data quality is not reduced so much. On the other hand, in the extreme case where the node is only powered on once, with a resulting duty cycle of 0.03, the R${}^{2}$ worsens approximately by 0.10 R${}^{2}$ in the case of the 20001 O${}_{3}$ sensor, and in the case of the 20001 NO sensor by 0.34 R${}^{2}$. In addition, Table 5 shows the results obtained for the same experiment but using the KNN and SVR as calibration models. It can be seen that these nonlinear models do not improve the MLR performance very much, since the sensors’ responses are very linear, except for the NO sensor where these nonlinear models are able to improve the calibration around 0.07 R${}^{2}$. Nevertheless, in terms of trend, the nonlinear models show the same decreasing trend of the R${}^{2}$ with the duty cycle since the impact of the duty cycle is on the representativeness of the data and does not depend on the machine learning model.

To summarize, the results show how, for the gas sensors analyzed and different sensor response times, the duty cycles obtained can vary. For example, assuming response times in the order of two minutes, duty cycles of 0.1 can be achieved, calibrating with hourly reference values and maintaining the quality of the calibrated data. Otherwise, duty cycles between 0.01 and 0.02 can be achieved if sensor response times are negligible.

## 6. Conclusions

In this article, we have studied the implications of the sensor gathering process on low-cost sensors for air pollution monitoring. The sensor sampling strategy is very important in cases where these sensors are mounted on battery-based IoT nodes with power consumption constraints. To conduct the experiments, we have built two prototype Captor nodes that collect tropospheric ozone, nitrogen dioxide, and nitrogen monoxide at a high frequency. The results indicate a clear relationship between the sensor sampling strategy, the quality of the resulting air pollution estimation, and the node’s power consumption defined by its duty cycle. To assess the quality of the data, we have calibrated the sensors using multiple linear regression, support vector regression, and k-nearest neighbors. In the case of having a duty-cycled node with negligible sensor response times, uniform sampling resulted in good data quality with few samples regardless of the sampling period used. On the other hand, when the sensors have response times, which is a common case, only sequential sampling is possible, and the results show how the duty cycle can be reduced by seven to ten times, reducing the energy consumption by the same amount, maintaining good data quality. In addition, it has been observed that each type of sensor may require a different sampling frequency to obtain good data quality. Thus, in the practical case of a node’s design where different sensing technologies, with different sensor response times to the ones shown in this work, are used, it would be necessary to characterize the most appropriate duty cycle based on the designer’s needs, the sensors used, and how the sensors are calibrated. As future work, it would be interesting to minimize the duty cycle of a node, while achieving good data quality, using data-driven approaches based on sparse sensing using a tailored basis.

## Figures and Tables

**Figure 1 sensors-22-03964-f001:**
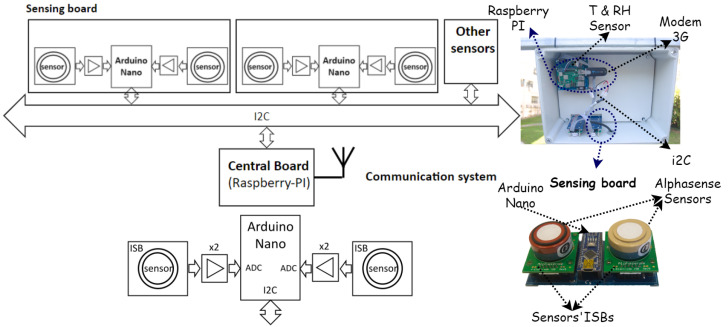
Air pollution data capture prototype Captor node on (**top**), and the Captor’s sensing board on (**bottom**). The prototype node has a Raspberry-based central processing unit connected to the sensing boards via an I2C bus. Each sensing board has an Arduino Nano microcontroller unit in charge of collecting the measurements from the electrochemical sensors and sending them to the Raspberry unit. The collected samples are then transmitted to the cloud via a 3G modem.

**Figure 2 sensors-22-03964-f002:**
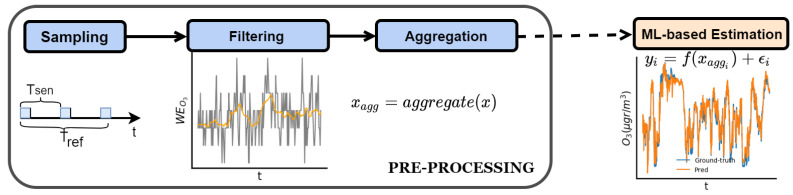
Sensor data gathering pipeline: from sensor sampling to machine learning estimation. First, sensors are sampled every $T_{\mathrm{sen}}$, then the samples collected during this period are filtered to eliminate possible outliers and aggregated to be sent to the cloud. There, the air pollution concentrations are estimated using the calibration models.

**Figure 3 sensors-22-03964-f003:**
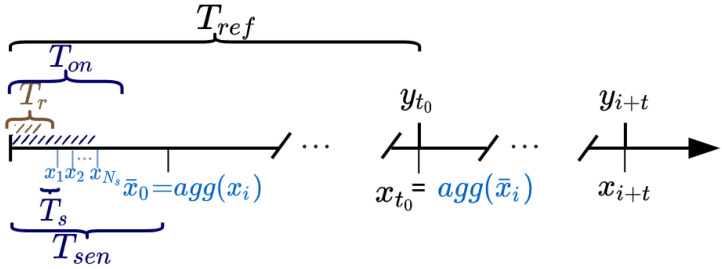
Sensor sampling scheme used throughout the article. $x_{i}$ are the sensor measurements while $y_{i}$ are the reference data measurements. Every $T_{\mathrm{sen}}$, the sensor is sampled by waiting for the sensor response time $T_{r}$ and collecting the $N_{s}$ samples to be aggregated. These $T_{\mathrm{sen}}$ samples can be further aggregated into $T_{\mathrm{ref}}$ to synchronize them with the reference values.

**Figure 4 sensors-22-03964-f004:**
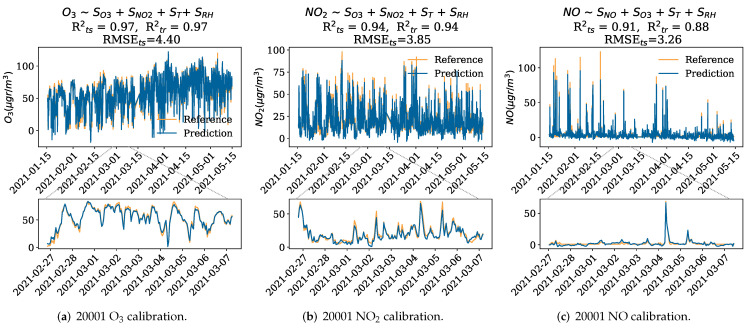
Calibration results for the O${}_{3}$, NO${}_{2}$ and NO sensor of Captor nodes 20001 and 20002. R${}_{\mathrm{ts}}^{2}$ and R${}_{\mathrm{tr}}^{2}$ denote the training and testing coefficient of determination, while RMSE${}_{\mathrm{ts}}$ denotes the root-mean-squared error for the testing using MLR. Bottom plots are a zoom of the plots above for a specific time interval.

**Figure 5 sensors-22-03964-f005:**
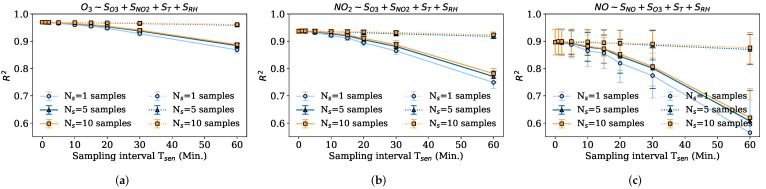
Average CV R${}^{2}$ and 95% confidence intervals for different sampling settings using MLR; sampling period $T_{\mathrm{sen}}$ and number of consecutive samples $N_{s}$ taken every period $T_{\mathrm{sen}}$. The solid lines denote the strategy that the $N_{s}$ samples are taken consecutively after a sensor response period, and the dotted lines denote the strategy that the $N_{s}$ samples are taken uniformly at $T_{\mathrm{sen}}$. (**a**) Captor 20001 O${}_{3}$ CV R${}^{2}$ for different $T_{\mathrm{sen}}$ and $N_{s}$. (**b**) Captor 20001 NO${}_{2}$ CV R${}^{2}$ for different $T_{\mathrm{sen}}$ and $N_{s}$; (**c**) Captor 20001 NO CV R${}^{2}$ for different $T_{\mathrm{sen}}$ and $N_{s}$.

**Figure 6 sensors-22-03964-f006:**
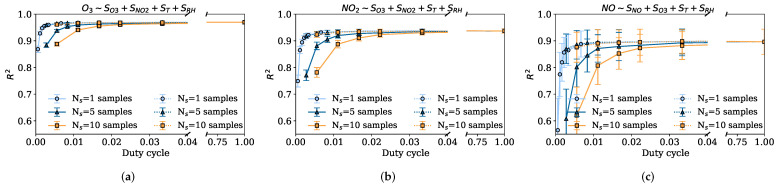
Average CV R${}^{2}$ and 95% confidence intervals for different duty cycles with negligible sensor response time ($T_{r}\;\approx \;0$) using MLR. Solid lines denote the strategy that samples the $N_{s}$ consecutively, and the dotted lines denote the strategy that samples the $N_{s}$ uniformly at $T_{\mathrm{sen}}$. (**a**) Captor 20001 O${}_{3}$ CV R${}^{2}$ for different duty cycles. (**b**) Captor 20001 NO${}_{2}$ CV R${}^{2}$ for different duty cycles. (**c**) Captor 20001 NO CV R${}^{2}$ for different duty cycles.

**Figure 7 sensors-22-03964-f007:**
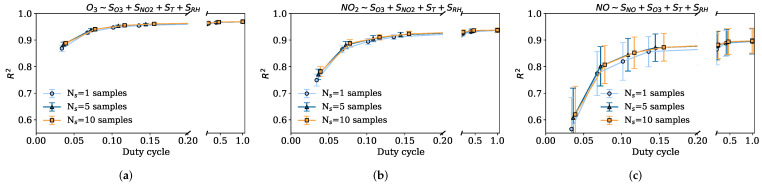
Average CV R${}^{2}$ and 95% confidence intervals for different duty cycles with a sensor response time equal to 2 min ($T_{r}\;=\;2\;\min$) using MLR. (**a**) Captor 20001 O${}_{3}$ CV R${}^{2}$ for different duty cycles. (**b**) Captor 20001 NO${}_{2}$ CV R${}^{2}$ for different duty cycles. (**c**) Captor 20001 NO CV R${}^{2}$ for different duty cycles.

**Table 1 sensors-22-03964-t001:** Sensor sampling periods used in the literature.

Work	Pollutants	Sampling Period ($T_{s}$)
Mijling et al. [7]	NO${}_{2}$	1 min
Sahu et al. [31]	O${}_{3}$, NO${}_{2}$	1 min
Ali et al. [17]	CO, NO${}_{2}$, PM	1 min
Becnel et al. [18]	CO, NO${}_{2}$, PM${}_{1}$ PM${}_{2.5}$, PM${}_{10}$	1 min
Nowack et al. [9]	NO${}_{2}$, PM${}_{10}$	30 s
Bigi et al. [10]	NO, NO${}_{2}$	20 s
De Vito et al. [11]	NO${}_{2}$, O${}_{3}$, NO	20 s
Si et al. [12]	PM${}_{2.5}$	6 s
Mead et al. [13]	NO	5 s
Han et al. [14]	O${}_{3}$, NO${}_{2}$, CO, SO${}_{2}$	2 s
Mead et al. [13]	CO, NO${}_{2}$	1 s
Astudillo et al. [15]	O${}_{3}$, CO	1 s

**Table 2 sensors-22-03964-t002:** Description of the datasets used for the experiments. $T_{s}$ is the sensor sampling period.

Node Label	Sensor	Deployment Period	$T_{s}$
*20001*	O${}_{3}$	2021/01/15–2021/05/15	2 s
	NO${}_{2}$	2021/01/15–2021/05/15	2 s
	NO	2021/01/15–2021/05/15	2 s
*20002*	O${}_{3}$	2021/01/15–2021/05/15	2 s
	NO${}_{2}$	2021/01/15–2021/05/15	2 s

**Table 3 sensors-22-03964-t003:** Sensor sampling parameters.

Parameter	Definition
$T_{s}$	Required time to take a sensor measure
$T_{\mathrm{sen}}$	Sensing node sampling period
$N_{s}$	Number of samples taken every sampling period
$T_{r}$	Sensor response time before valid measurements
$T_{\mathrm{ref}}$	Reference data period
DC	Sampling strategy duty cycle
$T_{\mathrm{on}}$	Time the microcontroller is switched on to collect sensor samples

**Table 4 sensors-22-03964-t004:** Machine learning calibration best subset of sensors found via forward stepwise feature selection.

Target Sensor	Best Subset
O${}_{3}$	O${}_{3}$, NO${}_{2}$, T, and RH
NO${}_{2}$	NO${}_{2}$, O${}_{3}$, T, and RH
NO	NO, O${}_{3}$, T, and RH

**Table 5 sensors-22-03964-t005:** Average CV R${}^{2}$ obtained with the multiple linear regression, k-nearest neighbors, and support vector regression models for the different sensors and duty cycles, with $T_{r}\;=\;2\;\min$.

Sensor	DC = 1.00			DC = 0.10			DC = 0.03		
	MLR	KNN	SVR	MLR	KNN	SVR	MLR	KNN	SVR
*20001* O${}_{3}$	0.97	0.96	0.98	0.95	0.94	0.96	0.87	0.86	0.88
*20001* NO${}_{2}$	0.94	0.94	0.96	0.89	0.90	0.92	0.75	0.77	0.78
*20001* NO	0.90	0.95	0.97	0.82	0.89	0.90	0.56	0.66	0.60
*20002* O${}_{3}$	0.97	0.96	0.98	0.94	0.93	0.95	0.84	0.84	0.85
*20002* NO${}_{2}$	0.92	0.92	0.94	0.88	0.89	0.91	0.74	0.76	0.78

## Data Availability

The Captors’ raw data used throughout the paper are openly available on Zenodo’s website [37].

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
