# Peer review of "Sampling Trade-Offs in Duty-Cycled Systems for Air Quality Low-Cost Sensors"

_sensors, 2022, doi:10.3390/s22103964_

Round 1

Reviewer 1 Report

The manuscript is well written and addresses an important issue, It can be considered for publication after the following queries are addressed.

  1. The authors have chosen electrochemical gas sensors  OX-B431 O3 sensors , NO2-B43F NO2 sensors, and NO-B4 NO 151 sensors for their study , it would be better if the authors elaborate with reasoning for choosing these sensors  which is used for sensing O3, NO2, and NO for their study. Air pollution can be caused by different gas species and moreover, in the title it mentions is mentioned as "air pollution". It would be better if the authors elaborate on choosing these sensors for the current study in the appropriate section in the manuscript.
  2.  Since in a mixed environment predicting the gases is a challenging task, can the current data-driven approach (model/ML algorithm/data sets) discussed here will be able to address this issue?

Author Response

Answer to Reviewer 1:
The manuscript is well written and addresses an important issue, It can be considered for publication after the following queries are addressed.

1. The authors have chosen electrochemical gas sensors  OX-B431 O3 sensors , NO2-B43F NO2 sensors, and NO-B4 NO sensors for their study, it would be better if the authors elaborate with reasoning for choosing these sensors  which is used for sensing O3, NO2, and NO for their study. Air pollution can be caused by different gas species and moreover, in the title it mentions is mentioned as "air pollution". It would be better if the authors elaborate on choosing these sensors for the current study in the appropriate section in the manuscript.

In this paper, we use a prototype data collection node (Captor node) that includes electrochemical sensors for O3, NO2, and NO. There is an extensive literature where O3, NO2 and NO sensors are mounted for air quality monitoring in urban areas. In addition, in this specific case, we use electrochemical sensors (alphasense manufacturer) which are affordable sensors for air quality monitoring. Moreover, this type of sensor has been the most common choice in recent years for low-cost air pollution monitoring. There are a variety of articles using electrochemical sensors of these gas species, which are cited in the article [7,9,11,13,31]. Following the reviewer's recommendation, in section 3.2 (low-cost sensors), line 151, we have mentioned that this electrochemical sensing technology has recently been widely used for air quality monitoring, and we have cited the corresponding articles.

2. Since in a mixed environment predicting the gases is a challenging task, can the current data-driven approach (model/ML algorithm/data sets) discussed here will be able to address this issue?

Yes, the calibration process is just that: predicting a gas from the measurements of a set of low-cost sensors which can include sensors measuring related gases. We study the impact of the sensor sampling (important post-processing step) on the quality of the calibration and its impact on the duty cycle. To define the calibration model, given that the sensing node has up to three sensors of three different gas species, we apply a forward feature selection to see whether the introduction of other of the two remaining sensors improves the calibration of a specific sensor. Thus, we can take advantage of the cross-sensitivities and correlations present between the different air pollutants. Thus, we can observe how to calibrate the NO sensor, introducing the O3 sensor (which measures both O3 and NO2) improves the estimation of this pollutant. This type of calibration where a sensor array of different pollutants is used is a technique commonly used in the calibration of low-cost sensors [10,25].

In accordance with the reviewer's recommendation, we have better specified the role of forward feature selection and the benefit of including more than one pollutant in the calibration of a sensor in section 5.1, line 355.

Reviewer 2 Report

This manuscript presented the influence of the sensor sampling strategy on low-cost sensors for air pollution monitoring. The relative simulation and machine learning data analysis works are performed and detail discussed on the O3, NO2, and NO sensors for ambient air monitoring. Overall, this work is helpful for gas sensor applications in real situations.

This manuscript is suggested for publication in Sensors with minor issues,

  1. Need more introduction on the three applied in-situ calibration methods for sensor researchers out of the data science field.
  2. Since the existence of NO2 and NO in the air could be correlated, is it possible to integrate NO2 into the calculation function for NO to reduce the high variability of NO?

Author Response

Answer to Reviewer 2:
This manuscript presented the influence of the sensor sampling strategy on low-cost sensors for air pollution monitoring. The relative simulation and machine learning data analysis works are performed and detail discussed on the O3, NO2, and NO sensors for ambient air monitoring. Overall, this work is helpful for gas sensor applications in real situations. This manuscript is suggested for publication in Sensors with minor issues,

1. Need more introduction on the three applied in-situ calibration methods for sensor researchers out of the data science field.

According to the reviewer's request we have added more information about the machine learning models used for in-situ sensor calibration, section 4.2, lines 314-322. We have explained the nature of the three machine learning models used, as well as information about the three models and further references for further guidance.

2. Since the existence of NO2 and NO in the air could be correlated, is it possible to integrate NO2 into the calculation function for NO to reduce the high variability of NO?

In section 5.1 (machine learning-based calibration) of the results section, we define the calibration models by forward step-wise feature selection. Since the sensing node has up to three sensors measuring different pollutants, it is interesting to see if the correlations and cross-sensitivities between pollutants can be exploited by adding other sensors in the calibration of specific sensor. Therefore, in this feature selection procedure, we add the sensor that increases the R2 the most at a time. In the case of NO, the sensor that increases the quality of the calibration the most is the O3 sensor (which measures both O3 and NO2), so the NO2 sensor does not provide any better goodness-of-fit improvement of the in-situ calibration model. The variability in the R2 of the calibration depends on the calibration model and on the specific data of this data set. However, the average R2 obtained in the calibration is a representative metric for the goodness-of-fit of the calibration and shows good results regardless of the variability.

Following the reviewer's concern, we have emphasized the use of forward feature selection in section 5.1, line 355.

Round 2

Reviewer 1 Report

The authors have addressed all the comments and hence  can be accepted for publication.

Author Response

thanks for the review.